# Teledentistry and Forensic Odontology: Qualitative Study on the Capacity of Implementation

**DOI:** 10.3390/ijerph20196807

**Published:** 2023-09-22

**Authors:** Céline Sabourin, Nicolas Giraudeau, Eric Baccino, Frédéric Camarasa, Laurent Martrille, Camille Inquimbert

**Affiliations:** 1Department of Forensic Medicine Lapeyronie Hospital, University of Montpellier, 191 Av. du Doyen Gaston Giraud, CEDEX 05, 34295 Montpellier, France; e-baccino@chu-montpellier.fr (E.B.); f-camarasa@chu-montpellier.fr (F.C.); laurent.martrille@chu-montpellier.fr (L.M.); 2Dental Department, Montpellier University Hospital, University of Montpellier, 34080 Montpellier, France; nicolas.giraudeau@umontpellier.fr (N.G.); camille.inquimbert@umontpellier.fr (C.I.); 3CEPEL, CNRS, University of Montpellier, 34060 Montpellier, France; 4EDPFM, Department of Legal Medicine, University of Montpellier, CHU Montpellier, 34000 Montpellier, France; 5IDESP, UMR UA11 INSERM, University of Montpellier, 34090 Montpellier, France

**Keywords:** forensic dentistry, telemedicine, identification, teledentistry

## Abstract

The postmortem identification of people without an identity is performed either by using DNA, dental charts, or fingerprints (until advanced decomposition prevents their study). The lack of forensic dentists who can conduct identifications lead us to reflect on the use of digital technology in this area. The aim of this study was to validate the organizational capacity of using teledentistry for the identification of bodies in a forensic medicine department. A mixed observational study was conducted on 55 cases between July 2020 and February 2021 in the Forensic Medicine and Thanatology Department of Montpellier University Hospital. The protocol was structured in five steps: an initial interview with the agent (a forensic autopsy technician/caregiver specialized in forensic medicine), regarding the idea they had in terms of using telemedicine in their daily practice; agent training in the telemedicine system; realization of a clinical examination using an intra-oral camera by the agent; data analysis by a dental surgeon; final interview with the agent. The study was conducted on 55 subjects. The average age was 54 years old, with more than two-thirds of the patients being male (69%). The videos had an average duration of 29 min. There was an increase in visit duration when rigidity was high; this was also the case when there were many dental specificities in the oral cavity. The quality of the videos was either good or excellent. This study showed that remote identification could be considered as a new non-invasive identification tool. Many features were analyzed to create a training guide for forensic institutes.

## 1. Introduction

Forensic odontology is a specialized discipline of forensic medicine that focuses on the study of the teeth and jawbone in any context. Particularly in the context of body identification, the teeth and oral system are of paramount importance. Teeth are a strong individual marker that resist extreme conditions (charring, immersion, putrefaction) [1,2,3].

The use of digital technologies is increasingly seen as essential to improve access, continuity, and quality of healthcare and to strengthen healthcare systems [4].

Telemedicine is described in France as a form of remote medical practice using information and communication technologies [5,6]. The objectives of this practice are numerous: to facilitate access to care, to improve the quality of life of patients, and to promote exchanges and coordination between the various healthcare actors.

Since 2014, Montpellier University Hospital has developed the e-DENT program [7] which allows asynchronous remote oral check-ups, especially for people with specific needs. The healthcare professional records specific information regarding the patient, required for an oral assessment by a dental surgeon. The latter performs an oral assessment remotely thanks to the information sent. The information includes patient identification, medical record elements, and, in particular, videos of the oral cavity taken with a specific intra-oral camera using fluorescent light (SoproCare^®^, ActéonGroup, Mérignac, France). A specific telemedicine software is used to simplify the collection and secure the transfer and storage of data (e-DENT^®^, Conex Santé, Labège, France). This practice has proven to be effective and reliable for remote dental screening, diagnosis, consultation, and treatment planning. There is growing evidence of the benefits of teledentistry in addressing public health needs and healthcare worker shortages. Several studies have reported the effectiveness, reliability, and validity of teledentistry for prevention, health promotion, and diagnostic accuracy of oral diseases, and their results are comparable to conventional examinations [8,9].

For several years, we have been interested in the use of teledentistry in forensic odontology. We call this new practice “forensic tele-odontology”. Very few studies are devoted to this innovative specialty [10,11], and yet many specific cases exist; for example, if the expert in forensic odontology, a key player in the identification protocol, and the one who is performing the postmortem clinical examination, cannot be present within a satisfactory period of time for many reasons, such as a medical desert (heterogeneous distribution of expert odontologists in France), mass disaster, war zone, or contaminated body. Dental evaluation can be crucial in the identification of missing persons and in forensic cases, such as major disasters and accidents. Even today, in criminal cases, personal identification is performed by traditional visual comparisons of antemortem dental records, when they can be recovered, and those obtained postmortem [12]. In 2021, we put forward the possibility of using forensic tele-odontology to perform a quality postmortem oral clinical examination [13,14]. Following this first article, it seemed essential to analyze the capacity of implementation of this type of practice in a forensic medicine department. We, therefore, launched a qualitative study on the organizational aspects of forensic tele-odontology.

The main objective of this work was to analyze the capacity to set up identifications by tele-legal dentistry by a non-odontologist health professional, and a remote dental surgeon in the legal medicine department of Montpellier University Hospital. The funeral chamber specificity caregiver work includes preparing the autopsy suite, moving bodies, and assisting the forensic doctor with various exam parts. This could involve weighing organs and collecting toxicology samples. Other responsibilities might include taking notes, photographing the body, or suturing to close a body. Therefore, the secondary objectives were to evaluate the obstacles and facilitators regarding the implementation of this practice within a forensic medicine department, to evaluate the time required for the amphitheater agent, to evaluate the learning curve of the forensic autopsy technician, and finally, to test the agent’s experience in using and implementing the practice.

## 2. Materials and Methods

### 2.1. Trial Design

This was a cross-sectional observational study of clinical tool validation on deceased subjects from the forensic medicine and thanatology department of Montpellier University Hospital. This mixed-method study was conducted from July 2020 to February 2021.

The study protocol was approved by the ethics committee according to French standards (Institutional Review Board IRB CHU Montpellier) under the number: 2019_IRB-MTP_05-04. The study was conducted in accordance with the Declaration of Helsinki.

### 2.2. Protocol

The study protocol was set up using 5 main steps (see workflow in Figure 1): (1) Initial interview of the amphitheater agent by a third party to assess their opinion of the project, their fears, and their expectations regarding the use of the device; (2) training of the agent by the expert dental surgeon in the basics of dentistry and the use of the teledentistry device. Training was conducted in two stages, namely a theoretical stage (demonstration of the software and the camera) and then a practical stage on two “test” bodies; (3) recording of the data necessary for the identification by tele-legal dentistry by the agent (non-removal of the maxilla; therefore, a non-invasive and practicable practice); (4) remote data analysis and assessment by the dental surgeon of the dental specificities; (5) final interview (following the same pattern as the initial interview).

### 2.3. Study Outcomes

We wanted to evaluate the non-odontologist professional’s motivation and possible preconceptions through an interview at the beginning of the study. The interview guide was constructed inductively and followed the framework used in accommodation facilities for the dependent elderly (Etablissements d’Hébergement pour Personnes Agées Dépendantes (EHPADS)) [15].

The elements studied for the practical part of the videos were video capture speed, the clinical quantification of rigor mortis (RM) (Table 1a), the number of dental specificities (Table 1b), and the quality of the videos ranked from 0 to 2 (0 being insufficient, 1 being good, and 2 being excellent).

A final interview was conducted to obtain the professional’s feelings, as well as the technical and administrative set-up before and after the video collection, and the necessary material.

### 2.4. Patients Inclusion Criteria

Our inclusion criteria were deceased subjects, aged over 18 years old, having stayed in the Forensic Medicine and Thanatology Department of the Montpellier University Hospital, autopsied or not. We excluded minor patients arriving in the same department.

### 2.5. Study Intervention

In addition to the classical equipment used for an autopsy, we used the e-DENT^®^ device (Conex Santé, Labège, France) composed of an intra-oral camera Soprocare^®^ (Acteongroup, Mérignac, France) associated with the e-DENT^®^ software (Version 3.1) to make the video recordings. The data transmitted on the server are secure, and the storage respects the compulsory standards of the “Agence du Numérique en Santé”. The dentist was asynchronously and remotely connected to the secure server through the telemedicine platform and was able to fill in the same clinical file to establish the postmortem odontogram. All diagnoses and reports were integrated into the e-DENT^®^ software for each patient.

### 2.6. Statistical Analysis

The data were collected using a secure software with an anonymity number (allocation of a random number) and then the data were transcribed into an Excel file. Quantitative variables were described by their means, and qualitative variables by their percentages and distributions. Trend lines were created by applying smoothing splines to the durations of the procedures, to illustrate trends in existing data and future predictions. All statistical analyses were performed using Microsoft^®^ Excel^®^ 2019 MSO (Version 2308 Build 16.0.16731.20052) 32 bit. Only descriptive statistics were performed.

## 3. Results

### 3.1. Patient Characteristics

The study was conducted on 55 patients in the Forensic Medicine and Thanatology Department of Montpellier University Hospital between July 2020 and February 2021. Our final sample was composed of 17 women (31%) and 38 men (69%), aged from 20 to 85 years with an average age of 54 years (Table 2).

### 3.2. Initial Interview

The non-odontologist professional announced that he had little experience in the dental field. His expectations of the training were clear: “the training should be quick, concise, and the software should be intuitive.” Concerns were raised about computer details: “I’m a little worried about the logins, the network connection, and if it will work in all the rooms.”

Regarding the equipment to be used in addition to the usual equipment: “it should be ergonomic and above all, not cause injury to the patient”, and “it should be easy to maintain, easy to store and efficient”. The importance of the study was emphasized: “I think that this study could bring a real valorization of our profession by providing new knowledge”; “I am very enthusiastic, on the one hand, to enrich my knowledge, and on the other hand to be able to help the other mortuary rooms”.

### 3.3. The Videos

The duration of the video recordings on average is 29.7 min (sd = 7.3); the majority of videos were made in 25 min (25.5%) (Figure 2).

The two longest videos (45 min) were made at the beginning of the study and the two shortest videos (15 min) were made toward the end of the study. The duration of the videos decreased over time from 33 min to 27 min on average. The dashed trend curve indicates a decrease in video duration with time and, therefore, experience. The x-axis reports the identification number progressively assigned to each patient, consecutively enrolled in the study: 1 is the first video recorded; 55 the last (Figure 3).

Here, 45% of our subjects had a rigor mortis score of 2, 3, or 4 and 55% had a score between 0 and 1. The more advanced the rigor mortis, the longer the duration of the visits. We can see that the duration of the videos was never less than 25 min for patients with RM evaluated at 3 or 4 (Figure 4).

Patients had an average of 12 dental specificities (sd = 6.7). The greater the number of dental specificities, the longer the video. Here, every point represents one video, and for each we can see the duration and the number of dental specificities found. We can see here that for the 40 or 45 min videos there are at the very least eight dental specificities. (Figure 5).

The videos were viewed by the dentist remotely. As a result, the remote dentist has never been in contact or seen the persons to be identified. The role of the dentist was to confirm that he had access to all the oral areas necessary to make a dental diagram and, thus, make a remote identification. He also had to assess the progress of the amphitheater agent. Finally, he had to qualify the videos by three characteristics according to their quality and the possibility/difficulty of making this diagram: insufficient (quality too low to make a diagnosis), good (sufficient quality that could be improved, due to parasitic movements affecting the fluidity of the image), and. finally, excellent (video with little or no undesirable movements). Identification is possible when the video is good or excellent. When the quality of the video was described as excellent, there were no shortcomings. There was very little need to go back and revise parts of it. The quality analysis of the videos was good in 78.2% of the cases and the quality was excellent for 21.8% of the cases. The excellent ones “appear” in November 2020, 3 months after the beginning of the recording. In our study, videos have always been of sufficient quality to be able to diagnose; none of the videos were insufficient; all dental specificities could be diagnosed by the remote dentist.

### 3.4. Final Interview

The non-odontologist professional did not change his opinion during the study: “I still think it would add value to our profession and I find the purpose very interesting, i.e., to have the possibility of identifying unidentified people through an electronic device.”

Motivation did not decrease over time and the feeling of progression in terms of dexterity was noted: “I felt much more comfortable during the last videos because I noticed that I could use both hands” and “I realized that each position was very important”. It is important to receive coaching on the first few recordings. As far as the equipment was concerned, the intra-oral camera was suitable. The computer equipment used was described as ergonomic, fast, and very practical. The recordings could be made at the beginning of, during, or at the end of an autopsy.

An important element was theoretical training: “I really think that initial dental theoretical training is very important and adding teeth cleaning would be a huge plus.” When the videographic data was collected, the set-up time was defined as quick, about 5 min, and the duration of the videos “will vary a lot depending on the type of patient”.

The non-odontologist professional has defined a list of equipment necessary for the proper functioning of the video data collection protocol. For the implementation, Wi-Fi access in the service, a room with electrical outlets and ceiling lighting (large enough to place the patient and equipment; temperature between 14 °C and 18 °C), a table (minimal length of 1 m), and a surgical site are required. Regarding the professional and its preparation, a single-use plastic apron, pair of single-use gloves, and protective mask are needed. When recording video, the intra-oral camera, a computer with the software and all associated connectivity, three different spacers numbered 1, 2, and 3 (Figure 6), a mouth mirror, toothbrush, and disposable bean are required. For the end of the examination, toilet, and preparation of the deceased the following is needed: the sheets must be destroyed, and soapy water, detergent and disinfectant, and moisturizer are required, as well as curved needle for mouth stitching (Figure 5), and a hairbrush.

The following synthetic protocol was used in the study. (1) Installation and preparation: dress preparation in accordance with recommend hygiene standards; preparation of the hardware, operator, and patient positioning. (2) Quick dental record on paper. (3) Video recording: installing a spacer to open the mouth cavity, carefully disengaging both maxillae if rigor mortis is important, performing a dry wash, referring to the dental record on paper, and starting filming. When video is being undertaken for an area, spacers should be positioned on the opposite sectors, and it is possible to use the mouth mirror as a “guide” for the camera and, thus, “slide” it so that there is less bumping. (4) Examination completion, washroom, and preparation of deceased: realization of the mouth stitch, washing the face, and massaging the patient’s face with moisturizer to reduce different reliefs that may have been left by the manipulations.

## 4. Discussion

### 4.1. Study Population

Among the 55 patients, 69% were male. We had a heterogeneous study population with differences in body stiffness, cause of death, state of decomposition of the body, and body damage.

### 4.2. The Videos

All videos have been analyzed as “good” or “excellent”, so we can say that the caregiver can pass the intraoral camera alone and allow identification by the expert dentist remotely. The learning curve evolved in a positive way. The most important improvement phase is between the 43rd and the last patient (55th), and more than half of the videos are rated as excellent (54%). We have highlighted the improvement in the quality of videos over time, and, therefore, the experience, when collected by the same individual. This result was found in another study conducted in 2020 [8,9].

We realized that the overall duration of the visits would evolve more as a function of the number of dental specificities found, rather than as a function of time. When collecting data videographics, there will be more details to highlight, and it will be necessary to focus on teeth having been treated to allow a correct diagnosis of the expert dental surgeon when he will visualize the data. The rigor mortis also influences the duration of the videos. During the collection of vestibular data, it is necessary to remove the jugal tissues, which is very complicated when rigor mortis is advanced.

Theoretical training is necessary for the operator to find their way around the oral cavity and to fill in the data correctly in the software. Practical training in the use of the camera is essential.

The software was appreciated for its intuitive use. We had made several modifications to adapt it to forensic dentistry.

From the third patient onward, our professional found exactly which equipment to use, and how to position themselves, and then continued in this way. Concerning the equipment imposed during the study, i.e., the Soprocare^®^ intra-oral camera, its main assets are its relatively small size and its lightweight nature, which make it a very practical tool. It is relatively easy to use after appropriate training and is non-invasive. As for the storage associated with the camera, the transport case is very practical thanks to its wheels, and can be easily stored in a closet. The internal storage boxes are optimized and are largely sufficient to store the plastic protections of the head of the camera, as well as all the connectivity equipment.

During practical training in the use of the camera, the U technique was taught, as follows: vestibular (1), occlusal (2), and, finally, palatal/lingual tooth by tooth from distal to mesial (3). (Figure 7) We had mentioned the possibility of using the second way (vestibular way of all of the teeth, then the occlusal part, and, finally, the palatal/lingual part) (Figure 8), but the first was privileged. The first technique caused a lot of problems because of the many changes in camera position. The second technique allowed less movement.

### 4.3. Teledentistry

Personal identification using dental evidence is decisive in the case of unidentified bodies. The contribution of forensic dentistry sometimes proves to be the only way of obtaining an identification, thanks to a comparative analysis. But the identification process of any unidentified human remains must respect best-standards in forensics and should always include the collection of all identifying postmortem data.

A systematic review on the development of digital technology in forensic dentistry revealed that no methods using digital technology have been accepted worldwide because of negative factors, such as expensive equipment and the cost of components [11].

Nevertheless, the protection of privacy and ethical considerations are paramount when processing personally identifiable information [16]. It is important to build up a database of dental images and recordings obtained during routine dental examinations. It has been suggested that a global dental database be created for personal identification from dental images, using teledentistry [17].

For a long time, documentation in forensic odontology was limited to photographs or 2D X-rays.

Many recent articles are based on “Computed Tomography” examinations [18,19]. Digital radiology and computer tomography has been shown to be important both in common criminalistics practices and in multiple fatality incidents [20].

A study carried out in 2021 put into practice a virtual dental autopsy and the remote assessment of forensic odontology.

Artificial intelligence is already being used to identify teeth and make diagnoses based on orthopantomography [21,22,23].

### 4.4. The Protocol

The protocol used allows to respect the recommended hygiene standards and respect the body of the deceased. It could, however, be improved by adding a brushing and complete cleaning of the patient’s oral cavity before any analysis. In the protocol used in the study, a “dry” cleaning was performed if there was too much fluid or substances covering the teeth. In a complete cleaning scenario, it would be a question of adding water and suction to really see more clearly and to clarify the areas of doubt related to the diagnosis.

For the implementation it will be necessary to have access to the Wi-Fi network in the service and a room with an electrical outlet and ceiling lighting. The room must be large enough (about 12 m^2^) to be able to arrange the patient’s body as well as the rest of the material, and, thus, to make the recordings in good conditions for both the professional and the body (the temperature must be between 14 °C and 18 °C). A table spacious enough to arrange the elements that will be used is also required, with a wingspan at least 1 m long and 40 cm wide.

### 4.5. Importance of Quality Dental Record Keeping

When comparing postmortem identification in odontology, as the name suggests, it is necessary to compare ante- and postmortem data. If there is no antemortem record, the identification task becomes increasingly complex, if not impossible. As such, the quality of record keeping of patients in dentistry, whether in private or public practice, is mandatory. Proper record keeping (photographs, X-rays, scanner, complete dental chart, clear care follow-up and precise) is necessary in daily practice but becomes indispensable in the framework of a judicial inquiry. This can only be achieved through the systematic implementation and updating of patient dental charts.

The question arises of setting up an odontogram database which, like the fingerprint database when it was created, could be synonymous with major technological and legal advances.

### 4.6. Criticisms and Limitations of the Study

We did not conclude that our results were significant, despite a relatively large number of subjects. We especially wanted to have trends in our results because this is the first study of its kind, which is why there are only descriptive statistics. But it would be very interesting to detail our statistics soon.

We also could add some criteria and axes of research in the future, like to differentiate several body states, i.e., fresh body, moderately altered bodies (first signs of putrefaction), rotten bodies, carbonized bodies, mummified bodies, degraded bodies (high velocity accidents, gunshot wounds, etc.), as well as weight (access and handling are more difficult in people with large builds), and, of course, the postmortem delay.

The study was based on the evaluation of a single non-odontologist operator, with a high level of motivation that may not be representative of all non-odontologist health professionals.

The quality of the videos, too, was measured subjectively by a single dental surgeon. It would surely be very interesting in a future study, to know if this competence would interest other care assistants, specifically those in the mortuary room. If this is the case, our training guide should be supported, as it is currently an informal draft. This guide could also contain more information about the financial investment of including this protocol in an institute or forensic medicine service.

## 5. Conclusions

This study shows us that forensic tele-odontology can be considered as a new non-invasive identification tool. This study allowed us to develop an informal training guide for forensic institutes and for caregivers specialized in forensic medicine who would like to train in teledentistry. Such a training guide could be of great help when the conditions do not allow doctors to bring an odontologist to the institute of forensic medicine directly and to quickly perform a gold-standard clinical examination. Forensic tele-odontology offers better working conditions and the obtention of an accurate postmortem odontogram, allowing the deceased person’s integrity to be preserved due to the non-removal of the maxillae. The construction of a global dental database for personal identification from dental images, using telemedicine, is a new challenge for researchers in dentistry and forensic dentistry.

## Figures and Tables

**Figure 1 ijerph-20-06807-f001:**
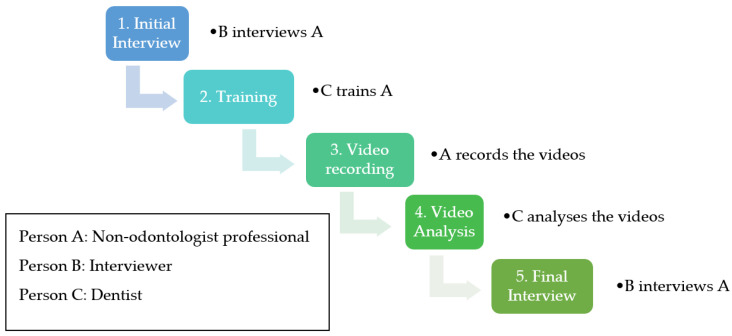
Workflow: the 5 steps of the study.

**Figure 2 ijerph-20-06807-f002:**
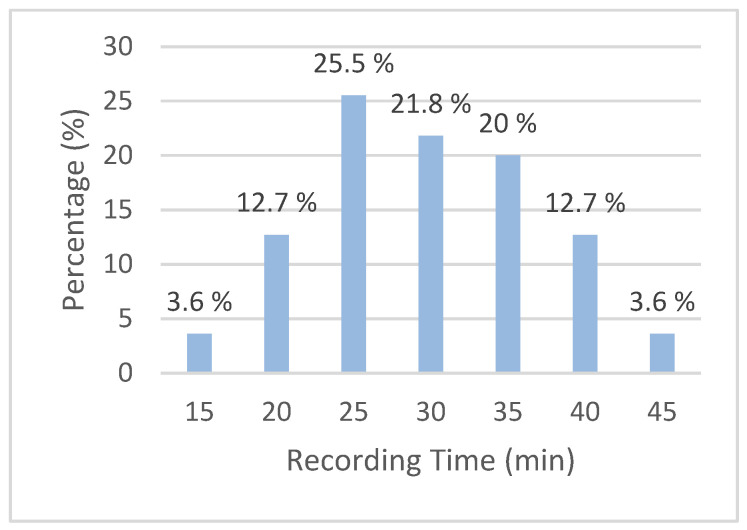
Duration distribution of videos in the study.

**Figure 3 ijerph-20-06807-f003:**
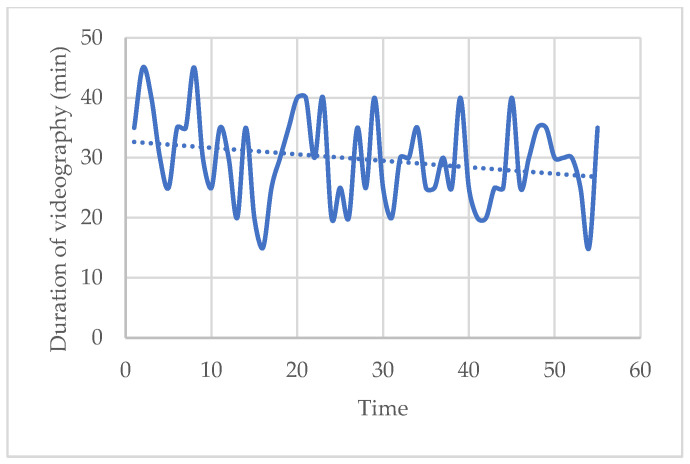
Trend of duration of videos according to patients as the study progresses.

**Figure 4 ijerph-20-06807-f004:**
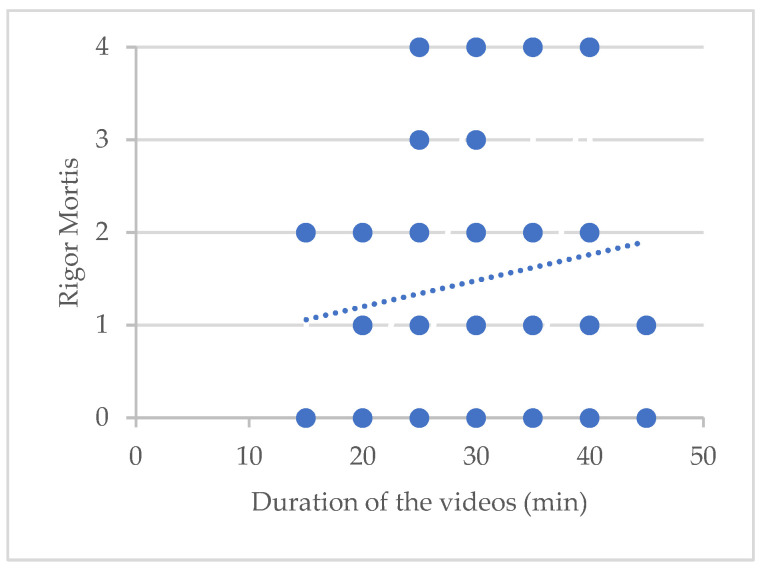
Trend in the relationship between visit duration and rigor mortis (RM): 0: no RM; 1: low RM; 2: moderate RM; 3: advanced RM; 4: very advanced.

**Figure 5 ijerph-20-06807-f005:**
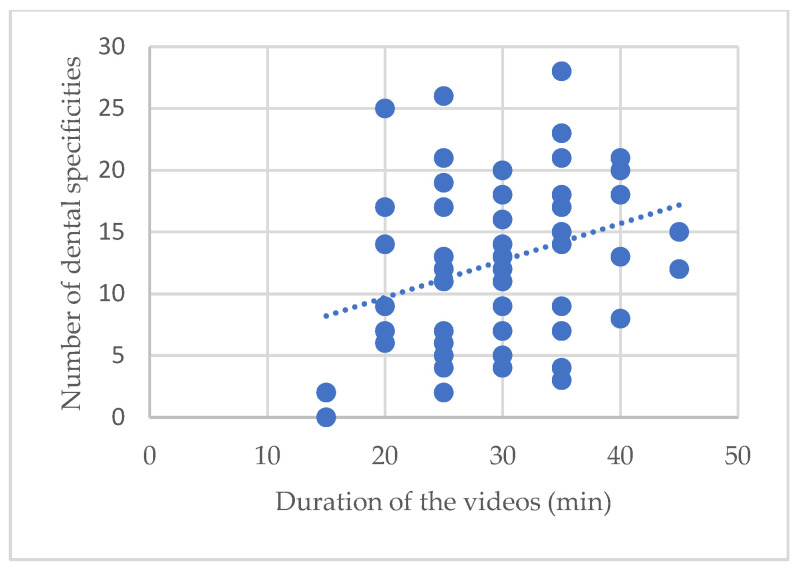
Trend in the relationship: duration of a video/number of total dental specifics.

**Figure 6 ijerph-20-06807-f006:**
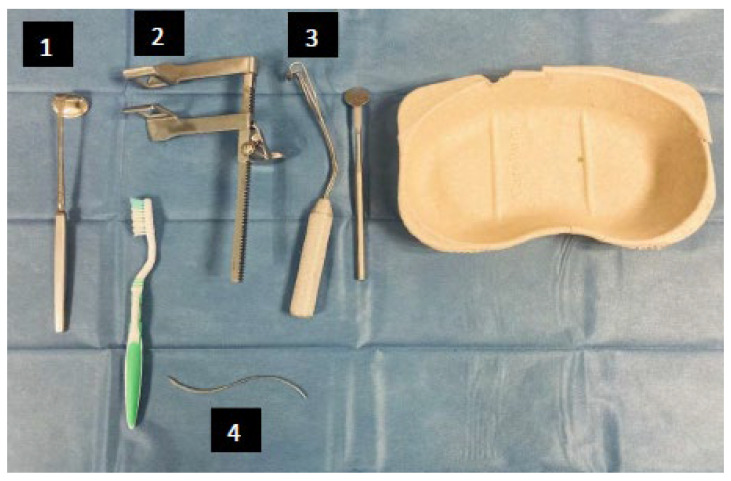
Main materials used by the professional in addition to the mandatory materials of the study.

**Figure 7 ijerph-20-06807-f007:**
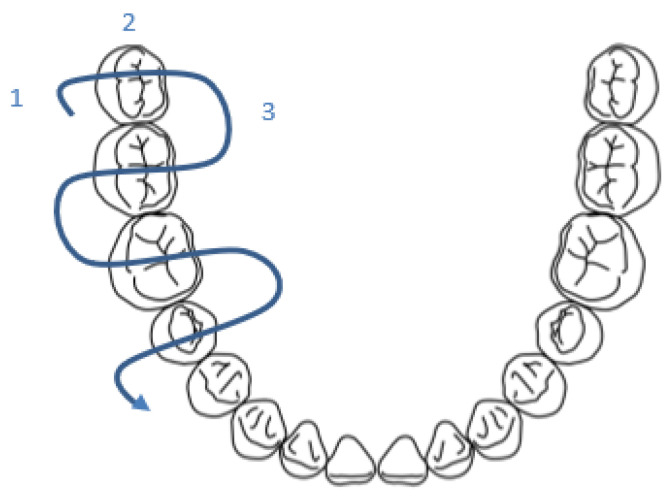
Representation of the technique chosen during the passage of the intra-oral camera (U technique).

**Figure 8 ijerph-20-06807-f008:**
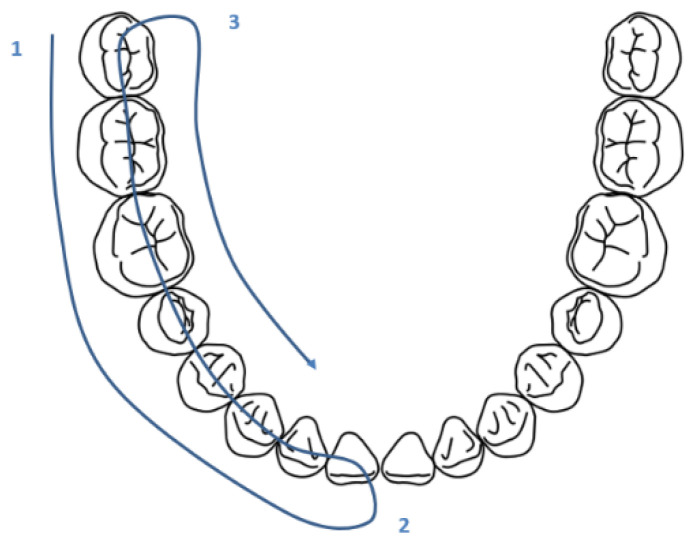
Representation of the second possible technique.

**Table 1 ijerph-20-06807-t001:** (**a**) Rigor mortis analyzed in forensic tele-odontology. (**b**) Dental specificities analyzed in forensic tele-odontology.

(a)
Rigor Mortis
0: Absence: No rigor mortis
1: Low: A weak press on the mandible allows the opening of the mouth, and the surgical retractor only maintains this position.
2: Moderate: The support is long and requires both hands, and the surgical retractor maintains the position but it is unnecessary to use the wheel.
3: Advanced: The support does not open the mouth enough but it is effective to slide the pallets of the surgical retractor; the key is used to release the mandible and maintain the position. Conventional spreaders can also be used.
4: Very advanced: The support with the hands is ineffective, and it is impossible to slide the pallets of the surgical retractor. Two retractors are used to release the mandible on both sides and then to slide the surgical retractor. Movements from top to bottom are applied using a simple retractor while turning the surgical key, which maintains the position.
**(b)**
**Dental Specificities**
Absence/presence of the tooth
No visible cavities
Covered wells and cracks
Presence of restoration without caries
Presence of restoration with caries
Ameliorative carious lesion
Dentin discoloration
Proximal involvement
Separate cavity
Marginal ridge fracture
Juxta-pulp caries
Root fragments
Fistula or abscess
Extracted tooth
Tooth to be extracted

**Table 2 ijerph-20-06807-t002:** Profile of study subjects.

Gender	Number (%)	Age (Year)
Male	38 (69%)	Average: 53
Female	17 (31%)	Average: 55
Total	55 (100%)	Average: 54
		Median (min; max): 54 (20; 85)

## Data Availability

Data available on request due to privacy or ethical restrictions. The data presented in this study are available on request from the corresponding author. Data are not available to the public due to their personal nature.

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
