# Peer review of "Teledentistry and Forensic Odontology: Qualitative Study on the Capacity of Implementation"

_ijerph, 2023, doi:10.3390/ijerph20196807_

Round 1

Reviewer 1 Report

Dear Authors,

Thank you for the opportunity to review the manuscript entitled "Teledentistry and forensic odontology: qualitative study on the capacity of implementation."

Although the paper is well-written, I have one or two concerns.  Please see the following comments:

1. Title: Upon first reading the title, I expect the paper to focus more on the factors affecting implementation of teledentistry in a forensic context, i.e. speaking a bit more to the legal framework, costs and specifics related to equipment.  However, the paper actually deals with the method, and how it is perceived by the forensic professionals.  I would therefore suggest that the title is changed slightly to exclude the words "capacity of implementation" and replaced with something along the lines of "perception of forensic practitioners." 

2. Line 137: Rather state "55 descedents..." otherwise it sounds like they died in your facility.

3. The conclusion refers to a training guide.  Is this a formal of informal guide? Is is something that will be published to be used by forensic professionals? Although the paper speaks to many aspects related to using teledentistry, it is not a concise guide.  It may be well worth publishing such a reference.  I suggest that the current format of such a guide (i.e. informal or formal) be mentioned in the conclusion, to avoid confusing the manuscript with a training guide.  

The last comment related more to the use of language:

4. Line 317: "They....." Are you referring to the training manual? Or else, who are they? If indeed indicative of the guide, just replace with "Such a training guide...."

Author Response

Thank you very much for your help, all your comments improve our article, 

We hope our answers suit you, 

Reviewer 2 Report

The Authors make a study on the validation of techniques of telemedicine for the identification of human bodies in forensic contexts. The study consisted in the evaluation of 55 cases. The results showed that techniques of telemedicine are a useful non-invasive identification tool-

This is an important contribution for forensic odontology and the manuscript is suitable for this journal. Overall, the manuscript is noticeably clear, well-structured and well referenced. Manuscript can be accepted without further changes. However, there are few typos/ errors that need attention:

* Line 12. In addition to DNA and dental charts, fingerprints are also a primary method of postmortem identification of human bodies (until advanced decomposition prevents their study).

* Line 108. Rigor Mortis is abbreviated as RC, but in line 171 is abbreviated as RM.

* Line 108. Table X???

* In the manuscript, the list of references should be described following the “Instructions for Authors”.

* The “Author List” in front matter and the “Author Contributions” in lines 327–330 do not match and contain the names of different authors.

Author Response

Thank you very much for involvement to help us improve this article, 

We hope our answers will suit you, 

Reviewer 3 Report

MATERIALS & METHODS:

“EHPADS”: the meaning of this abbreviation must be reported the first time it is used.

Please add a description of each stage of rigor mortis: which was the difference between low, moderate etc.?

“Table X” should be “Table 2” or “table 1” should be divided in ”1a” and “1b”.

Exclusion criteria should be added.

Please check which version of Excel was used.

 “a first species error” should be “a type 1 error”… but actually it is not clear which kind of statistical tests were applied: it looks like only descriptive statistics was performed.

RESULTS:

“17 women and 38 men” please add also percentage values.

“with an average age of 54 years”: when age is investigated, the mean value is followed by its standard deviation and also the median value is reported.

“Time represents the duration of the study”: this definition is misleading. The x axis reports the identification number progressively assigned to each patient, consecutively enrolled in the study.

Figure 2 should be implemented: “Patient” should be used instead of “Time” and dots – instead of a continuous line - should be used to clearly identify each patient.

Figure 3: why recording time is reported from longer to shorter videos? Normally x-axis values increase progressively from left to right.

It looks like figures 4 and 5 were exchanged.

x-axis dots of figures 4 and 5 are not correctly positioned, as only values multiple of 5 were represented: it looks like real values were approximated to the closest number multiple of 5.    

“3 different spacers” should be “three different spacers”

CONCLUSIONS:

“to do a standard gold clinical examination” should be “to do a gold standard clinical examination”

Few typo errors

Author Response

Thank you very much for your help in improving our article, your comments are very interesting and relevant,

We hope our answers will suit you,
